# Adaptive Modulation and Coding for Underwater Acoustic Communications Based on Data-Driven Learning Algorithm

Lianyou Jing [1,2,*] , Chaofan Dong [1] , Chengbing He [3], Wentao Shi [3] and Hongxi Yin [1,2]

1   School of Information and Communication Engineering, Dalian University of Technology,
    Dalian 116024, China
2   The State Key Laboratory of Integrated Services Networks, Xidian University, Xi'an 710071, China
3   School of Marine Science and Technology, Northwestern Polytechnical University, Xi'an 710072, China
*   Correspondence: lyjing@dlut.edu.cn

**Abstract:** With the development of the underwater acoustic (UWA) adaptive communication system, energy-efficient transmission has become a critical topic in underwater acoustic (UWA) communications. Due to the unique characteristics of the underwater environment, the transmitter node will almost always have outdated channel state information (CSI), which results in low energy efficiency. In this paper, we take full advantage of bidirectional links and propose an adaptive modulation and coding (AMC) scheme that aims to maximize the long-term energy efficiency of a single link by jointly scheduling the coding rate, modulation order, and transmission power. Considering the complexity characteristics of UWA channels, we proposed a bit error ratio (BER) estimation method based on deep neural networks (DNN). The proposed network could realize channel estimation, feature extraction, and BER estimation by using a fixed pilot of the feedback link. Then, we design a channel classification method based on the estimated BERs of the modulation and coding scheme (MCS) and further model the UWA channels as a finite-state Markov chain (FSMC) with an unknown transition probability. Thus, we formulate the AMC problem as a Markov Decision Process (MDP) and solve it through a reinforcement learning framework. Considering the large state-action pairs, a double deep Q-network (DDQN) based scheme is proposed. Simulation results demonstrate that the proposed AMC scheme outperforms the fixed MCS with a perfect channel information state, and achieves near-optimal energy efficiency.

**Keywords:** underwater acoustic communications; adaptive modulation and coding; BER estimation; channel classification; double deep Q-network

## 1. Introduction

As a key mobile node for underwater observation and development, the underwater vehicle needs to continuously exchange information with other platforms or nodes to achieve information sharing. As an indispensable key technology in the development of underwater vehicles, underwater information transmission has become one of the most active and rapidly developing research fields in recent years [1]. Since underwater acoustics (UWA) communication is the only way to conduct reliable underwater communication at medium and long distances, it has always been the top priority of underwater communication research [2].

In order to achieve efficient underwater acoustic communication, reliable data communication links between underwater nodes need to be established. In practice, most underwater nodes are often powered by batteries. Since the nodes are deployed underwater, battery replacement is time-consuming and expensive. Therefore, long-term energy efficiency is an important criterion for evaluating the performance of a communication system.

Due to the unique characteristics of the underwater environment, UWA channels are usually regarded as one of the most challenging communication channels [3,4]. A distinctive

characteristic of UWA channels is the rapid time-variant, i.e., the channel state changes rapidly with time and location. This dynamic characteristic makes it very difficult to achieve long-term energy-efficient transmission with a fixed modulation. Thus, it is necessary to adopt a dynamic transmission strategy according to the channel state information.

The adaptive modulation and coding (AMC) technique tracks the channel dynamics and adaptively switches among a set of predefined modulation and coding schemes (MCSs) for the most efficient transmission. Compared with the traditional fixed MCS UWA communication system, the AMC system could achieve better energy efficiency. In an AMC system, the transmitter selects the optimal transmission rate or energy consumption while satisfying the bit error rate (BER) requirement. Thus, the transmitter needs to know the BERs of various MCSs under a certain UWA channel. For terrestrial wireless communications, the signal-to-noise ratio (SNR) is usually chosen as a metric directly related to BER [5,6]. However, unlike the wireless communication channels, the UWA channels are characterized by a long delay spread, fast time-variant, severe Doppler effect, and limited available bandwidth. Currently, there is no widely accepted mathematical model for UWA channels. Thus, it is hard to find the relationship between the channel parameters and BER performance using model-based methods. Besides, the time-variant of UWA channels is usually unknown, which also increases the difficulty of AMC UWA communications.

In this paper, we propose a data-driven approach for AMC UWA communications. The transmitter node first uses the received pilot signal of the feedback link to estimate the BERs of all MCSs. Then, the transmitter node classifies the feedback UWA channels based on the estimated BERs. According to the classification results and the own system state, the transmitter chooses the optimal MCS for the next transmission. The key contributions of this paper can be summarized as follows.

(1) We propose a BER estimation method based on a deep neural network (DNN). A fixed pilot signal is transmitted in each transmission, and the received pilot signal is directly used as the input of the DNN. The BER label of the network is also preprocessed to improve the performance. The proposed network conducts all the functions, which include channel estimation, feature extraction, and BER estimation.

(2) Different from existing works that classify the UWA channels using the channel features, we directly use the estimated BERs of all MCSs to classify the feedback UWA channel. A channel classification criterion based on the BERs of all MCSs is designed. Based on the channel classification method, we could model the UWA channels as a finite-state Markov chain (FSMC) with unknown transition probability.

(3) We formulate the adaptive modulation and coding problem as a Markov Decision Process (MDP) and solve it through a reinforcement learning (RL) framework. Considering that the transition probability of the channel state is unknown and the large state-action pairs, we propose a double deep Q-network (DDQN) based method to solve the problem.

(4) We use a statistical UWA channel model to generate complex-valued UWA channels to evaluate the proposed performance. Simulation results show that the proposed BER estimation method, channel classification method and DDQN-based AMC scheme have good performance. The proposed scheme achieves near-optimal energy efficiency with perfect channel information.

## 2. Related Works

The AMC communication is a hot-pot research point for UWA communications. In 2000, Stojanovic et al. proposed a simple adaptive modulation scheme for UWA communication [7]. The multipath, Doppler spread, and SNR measured by a wideband probe were used for selecting the best modulation from phase-coherent (PSK and QAM) and noncoherent (FSK) modulations. In Ref. [8], the channel capacity and post-equalization SNR were used for adaptation metrics. In Ref. [9], a variable-rate adaptive modulation system based on instantaneous SNR information was proposed. The channel fading effects were modeled as a Nakagami-*m* distribution, where the parameter *m* is estimated over a

whole experiment or over smaller time windows throughout the experiment, depending on the variability of the SNR. Stojanovic et al. proposed an adaptive orthogonal frequency division multiplexing (OFDM) modulation based on the predicted channel to maximize the system throughput under a target average BER [10]. Zhou et al. used the effective SNR after channel estimation and channel decoding for selecting the MCS scheme in the UWA OFDM system [11,12]. Ref. [13] proposed an adaptive design for OFDM transmission systems by utilizing the long-term stability of the second-order statistics of the channel state information (CSI). The analytical expression of signal-to-interference-plus-noise-ratio (SINR) at each subcarrier to reach the target error performance based on the statistical information of the CSI is derived. Based on the analytical expression, adaptive coding and a bit-power loading algorithm are proposed. This kind of method relies on the accuracy of CSI. The inaccurate CSI can lead to improper modulation schemes, which in turn leads to low communication efficiency. Ref. [14] proposed a DNN-based adaptive UWA OFDMA system for CSI prediction.

Recently, machine learning (ML) technology has found wide applications in broad fields, including image/audio processing, economics, and computational biology. Correspondingly, many intelligent products derived from deep neural networks have appeared. For instance, Refs. [15,16] implemented natural scene recognition and real-time face recognition on FPGA. For scenarios with strict energy efficiency requirements, using customized chips such as FPGAs can obtain stronger computing power and lower power consumption; in other words, high energy efficiency. The work of [17] has substantially improved the energy efficiency of the hand pose estimation algorithm. The FPGA-based NLP acceleration framework proposed by [18] has improvements in storage, performance, and energy efficiency. In Ref. [19], the authors realized a Stochastic Computing based LSTM in FPGA, which can reduce 73.24% energy cost compared to baseline binary LSTM. Moreover, there are also some interesting results obtained by introducing ML into the field of communications [20]. In Ref. [21,22], the authors considered more channel metrics, such as received SNR, output SNR, channel average fade duration, channel RMS delay spread, Doppler spread, and then proposed a decision tree-based method for adaptive modulation and coding scheme. In our previous works [23,24], six-dimensional features of UWA channels are considered for the UWA AMC scheme. This kind of work formulated the adaptive modulation and coding problem as a multi-class classification problem. A similar idea was suggested in [25,26], where different ML algorithms were adopted.

The above works only consider the instantaneous channel state and do not consider the correlated of the channel in time. In fact, the UWA channels are time correlated, and some works utilize the character. In Ref. [27], Wang et al. proposed an adaptive transmission scheme for a point-to-point UWA communication system. The channel within each epoch was characterized by a compound Nakagami-lognormal distribution, and the evolution of the distribution parameters was modeled as an unknown Markov process. Then, the adaptive transmission problem was formulated as MDP and solved based on the RL algorithm. In Ref. [28], an adaptive modulation based on the Dyna-Q algorithm was proposed for a single-input multiple-output (SIMO) acoustic communication system. The channel state was characterized by the effective signal-to-noise ratio (ESNR) after time-reversal processing and channel estimated based on decision feedback equalization. Ref. [29] proposed an AMC UWA communication based on RL. The Q-learning algorithm was adopted in the paper. In Ref. [30], the authors further proposed an RL-based AMC scheme for UWA image communication. In Ref. [31], an adaptive modulation for long-range UWA communication was proposed. The authors selected the optimal modulation from four pre-set modulations based on a prior evaluation of the channel. By training a classifier using an acoustic propagation model, the authors lock onto the channel's important features, thereby reducing the sensitivity to mismatches in the environmental information. In Ref. [32], the authors proposed a TAS-DQN approach to choose the optimal transmit frequency and power of a single link to achieve the best energy efficiency.

### 3. System Model

We consider a point-to-point UWA communication system consisting of a pair of transmitter and receiver. Both nodes could send signals. Before each transmission, the transmitter node has already received a feedback signal from the receiver node. The feedback signal contains the transmission result of the previous transmission. The feedback link usually adopts a low rate coding to make sure there is no error. Considering the dynamic characteristic of UWA channels, we model the UWA channels as an FSMC with unknown transitional probabilities. Thus, there exists a relationship between the feedback channel and the forward channel.

The transmitter node chooses an MCS for current transmission based on the feedback transmission result and its state. In this paper, the MCS includes the coding rate, modulation order, and transmission power. Then, the transmitter encodes, modulates, and transmits the information data according to the selected MCS. Here, we adopt single carrier frequency domain equalization (SC-FDE) modulation [33–36]. The SC-FDE system adds a cyclic prefix (CP) at the transmitter to convert the linear convolution into a cyclic convolution and uses FFT/IFFT to reduce the computational complexity. It has become a promising method to solve ISI and Doppler spread for UWA communications due to the advantages of lower PAPR and insensitivity to frequency offset. Undoubtedly, the proposed method can be extended to other communication schemes.

#### 3.1. Sc-Fde Modulation

For the $n$-th transmission, the information bit is first encoded with coding rate $r_n$, where $r_n$ is an element of a finite set of channel coding rates $\mathcal{R}_c = \{r_1, r_2, \cdots, r_I\}$. Then, the coded bit is interleaved and mapped to the $M_n$-order PSK symbol, where $M_n$ is also from a finite set of discrete modulation sizes $\mathcal{M} = \{M_1, M_2, \cdots, M_J\}$. Here, we ignore the influence of the pilot and CP on the communication rate. Thus, the total communication data rate is $B \cdot r_n \cdot \log_2 M_n$, where $B$ is the bandwidth. After adding CP, the signal is sent to the UWA channel through the transducer with transmission power $P_n$, where $P_n \in \mathcal{P}$, $\mathcal{P} = \{P_1, P_2, \cdots, P_K\}$.

Assume the channel is constant during one data block. Then, the received symbols can be written as

$$y = Hx + w, \tag{1}$$

where $x$ is the transmitted symbol vector containing the CP, and $H$ is a circulant channel matrix constructed by the channel impulse response $h$. $w$ represents the environmental noise, which is modeled as additive white Gaussian noise with a mean value of 0 and variance of $\sigma^2$. The frequency-domain representation of (1) is given as

$$\begin{aligned} Y = Fy &= FHF^H Fx + Fw \\ &= \Lambda X + W, \end{aligned} \tag{2}$$

where $F$ is the normalized FFT matrix. Since $H$ is a circulant channel matrix, $\Lambda$ is a diagonal matrix in which the diagonal elements are the frequency domain representation of the time-domain channel response.

For the SC-FDE system, the receiver first estimates the channel. In this paper, we adopt a least square (LS) based channel estimation method. Let $\hat{\Lambda}$ denote the diagonal matrix based on the estimated channel, then the MMSE detection of the received symbol is given by

$$\hat{X} = \frac{\hat{\Lambda}^*}{|\hat{\Lambda}|^2 + \sigma^2} Y. \tag{3}$$

Applying the IFFT to transform $\hat{X}$ into the time domain, the demodulated signal is given as

$$\hat{x} = F^H \hat{X}. \tag{4}$$

The receiver then makes a decision and decodes the information.

### 3.2. Underwater Acoustic Channel Model

Underwater acoustic channels are usually recognized as one of the most complex communication channels. The UWA channel gain is generally described as distance- and frequency-dependent large-scale fading and the time-varying multi-path fading. The average SNR at the receiver is calculated by the transmission power $P$, the noise level, and the path loss. Based on the Urick's model, the path loss over a distance $d$ (in km) and frequency $f$ (in kHz) is given by

$$L(d,f) = d^k a(f)^d, \tag{5}$$

where $k$ is the geometric spreading coefficient, which ranges from 2 to 4 and is usually set to 1.5, and $a(f)$ is the absorption coefficient. The path loss can be expressed in dB form,

$$10 \log L(d,f) = k \cdot 10 \log d + d \cdot 10 \log a(f). \tag{6}$$

The absorption coefficient $a(f)$ (in dB/km) can be estimated by using Thorp's formula [37],

$$10 \log a(f) = \frac{0.11 f^2}{1 + f^2} + \frac{44 f^2}{4100 + f^2} + 2.75 \times 10^{-4} f^2 + 0.003. \tag{7}$$

The power spectral density of the ambient noise (in dB re $\mu$Pa per Hz) is the sum of four basic sources as follows:

$$N(f) = N_{tu}(f) + N_w(f) + N_s(f) + N_{th}(f), \tag{8}$$

where $N_{tu}(f)$, $N_w(f)$, $N_s(f)$, and $N_{th}(f)$ are turbulence, wind driven waves shipping and thermal noise, respectively. The expressions of these four noises are given as

$$10 \log N_{tu}(f) = 17 - 30 \log f, \tag{8a}$$

$$10 \log N_w(f) = 50 + 7.5 w^{\frac{1}{2}} + 20 \log f - 40 \log(f + 0.4), \tag{8b}$$

$$10 \log N_s(f) = 40 + 20(s - \frac{1}{2}) + 26 \log f - 60 \log(f + 0.03), \tag{8c}$$

$$10 \log N_{th}(f) = 15 + 20 \log f, \tag{8d}$$

where $w$ represents the wind speed with the unit m/s, and $s$ denotes the shipping activity ranging from 0 to 1.

The transmission source level (in dB) is given by

$$SL = 170.8 + 10 \log P + 10 \log \eta, \tag{9}$$

where $\eta$ is the electro-acoustic conversion efficiency of the transducer. Then, the received SNR is calculated as

$$SNR = SL - 10 \log L(d,f) - 10 \log N(f) - 10 \log B + DI, \tag{10}$$

where $B$ is the signal bandwidth, DI is the directivity index and set to zero.

The above expression is only used to calculate the SNR of the receiver. In fact, the factor that affects the transmission performance is not only the SNR, but also the structure of the

channels. For example, the positional change of the transmitter node and the receiver node will lead to a change in the channel structure, which in turn causes the channel capacity to fluctuate greatly. Figure 1 gives the simulated BER result for different UWA channels. The UWA channels are simulated by a statistical model in [38], while the water depth is 100 m, and the depth of the transmitter node and receiver node varies from 20 m to 80 m. To obtain multiple channel structures to demonstrate their impact on communication performance, we move receiver nodes from 2500 m to 5000 m and normalize all channels. The modulation is QPSK with 1/2 convolutional code and the SC-FDE scheme is adopted. From this figure, the BER ranges from $10^{-2}$ to $10^{-4}$ for SNR = 14 dB. The dynamic range of the BER is very huge.

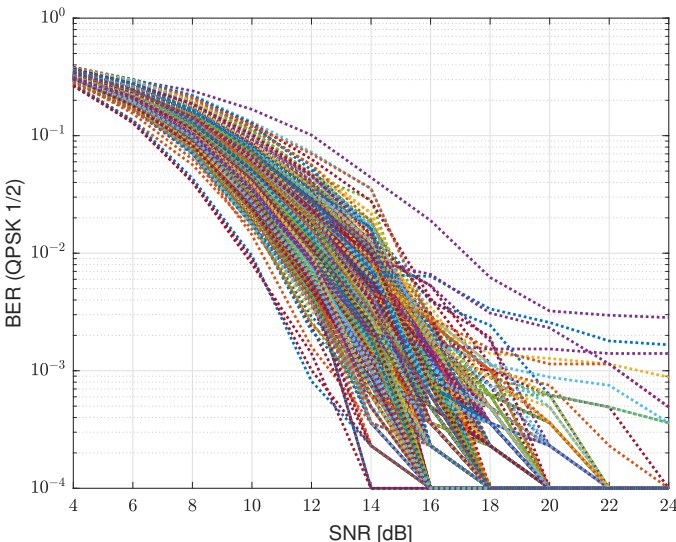

**Figure 1.** The BER performance for the same MCS system under different simulated UWA channels.

In addition, the time-variant of the UWA channel is also not considered in the above expression. In general, the complex marine dynamic process will make the water body non-uniform. Random fluctuations in the sea surface and unevenness in the seabed also affect the propagation of sound waves in the water. The speed of sound will also vary with water depth, temperature, salinity, etc. These factors lead to significant spatial differences and temporal fluctuations in the UWA channel, e.g., large Doppler spread. Furthermore, due to the slow propagation speed of sound waves in water, the mobile UWA communication system will incur long transmission delays, which may make the feedback channel state obsolete. This is unacceptable for the UWA communication system that directly depends on the channel state. To express the dynamic of UWA channels, in this paper, we model the UWA channels as an FSMC with unknown transitional probabilities. Assuming slowly time-varying and quasi-stationary channels, the channel state is stationary during each transmission and is allowed to change in the subsequent transmission according to the Markov model. Let $\boldsymbol{h}_i$ denote the channel gain in $i$-th transmission, which is quantized into $G$ levels. The state transition probabilities $P_{i,j}$ are unknown. Figure 2 shows the schematic diagram of the FSMC, which contains $G$ channel states.

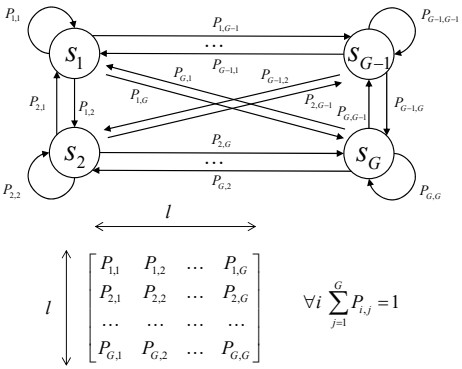

**Figure 2.** The schematic diagram of the FSMC, which contains $G$ states.

### 3.3. Problem Formulation

In this paper, our goal is to select the optimal MCS to maximize the packet delivery ratio and energy consumption, i.e., maximize the energy efficiency, under the constraint that the transmission rate and performance satisfy the communication demand. In a communication system, as we all know, BER performance is related to the channel, modulation, and coding scheme. In order to simplify the expression, we use the mode index $q_n = 1, 2, \cdots, Q, Q = I \cdot J$, to express different modulation orders and coding rates. Thus, the transmission parameters of the $n$-th transmission can be expressed by a vector $[P_n, q_n]$. Then, for the $n$-th transmission, the BER $e_n$ can be expressed as the function of $q_n$, $P_n$, and the channel $\boldsymbol{h}_n$,

$$e_n = \boldsymbol{f}(q_n, P_n, \boldsymbol{h}_n). \tag{11}$$

Let $\delta$ denote the maximum tolerable BER (i.e., quality of service (QoS) ) at the receiver. Then, if the BER is less than $\delta$, the transmission successes and the communication data rate $c_n$ are determined by the selected coding rate $r_n$ and modulation order $M_n$. Otherwise, the transmission failed and $c_n = 0$. Then,

$$c_n = \begin{cases} 0, & \text{if } e_n > \delta, \\ B \cdot r_n \cdot \log_2 M_n, & \text{if } e_n \leq \delta. \end{cases} \tag{12}$$

Besides, in order to make sure the communication link works well, we also set a minimum tolerable communication rate $c^*$. After a total of $N$ transmissions, the average transmission rate should not be less than $c^*$.

We aim to maximize energy efficiency by arranging the optimal transmission power and choosing the optimal modulation order and coding rate. Assume that the transmission duration $T$ in each transmission is fixed, then the transmission energy for $n$-th transmission is $E_n = P_n T$. Let $b_n$ present the transmitted data in $n$-th transmission, thus, the problem can be formulated as

$$\max_{P_n, r_n, M_n} \frac{\sum_{n=1}^{N} b_n}{\sum_{n=1}^{N} E_n} \tag{13}$$

$$\text{s.t. } \frac{1}{N} \sum_{n=1}^{N} c_n \geq c^*, \tag{14}$$

$$P_n \in \mathcal{P}, r_n \in \mathcal{R}_c, M_n \in \mathcal{M}. \tag{15}$$

The problem in (13) could be solved well if (11) has a definite mathematical expression. However, due to the lack of a mathematical model for the UWA channels, we do not have a mathematical expression for (11) right now.

In this paper, we propose a data-driven approach to realize the AMC UWA communication. Figure 3 shows the structure diagram of the proposed AMC system. At each

transmission, the receiver node first sends a feedback signal to the transmitter node. The feedback signal contains a fixed pilot signal and previous transmission result. The pilot is fixed for each transmission and is known for both the transmitter and receiver. At the transmitter node, the received feedback signal is first divided into the received pilot signal and the received transmission result. The received pilot signal contains information about the feedback UWA channels. Thus, we estimate the BERs of all MCSs ($q = 1, \cdots, Q$) based on the feedback pilot signal, and further classify the feedback channels. On the other hand, we obtain the previous transmission result by demodulating the feedback signal. Since we model the UWA channels as FSMC, we choose the optimal MCS for the current transmission by the DDQN algorithm according to the channel classification result of the feedback channel and the previous transmission result.

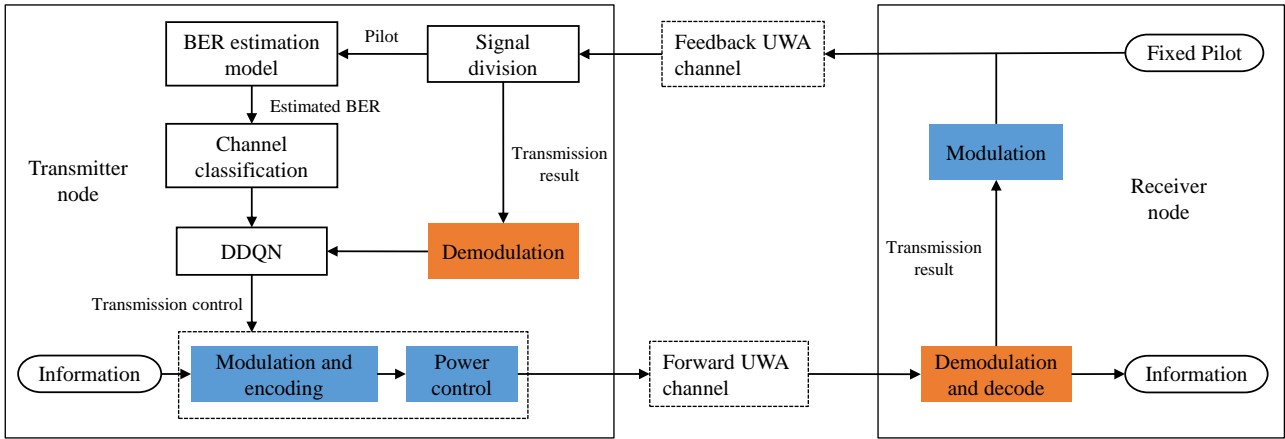

**Figure 3.** The structure diagram of the proposed AMC system.

## 4. Uwa Channel Classification

For the AMC scheme, the transmitter node first needs to obtain the UWA channel features and then chooses the optimal MCS. However, it is hard to directly use the channel state information due to the complexity of UWA channels, even the channel impulse response is well estimated. In this section, we introduce a new channel classification method based on the estimated BERs of all MCSs.

### 4.1. Channel Classification Based on BERs

As introduced in the above section, two factors affect the transmission performance of the UWA channels: one is the background noise level and the transmission loss, and the other one is the channel structure. The first factor affects the received SNR and further affects the transmission performance. The second one is that the doubly selective UWA channels introduce interference between transmitted symbols.

Most of the existing works classify the UWA channels based on the channel features. However, for a specific UWA channel $h$, there are too many features that may affect the performance, such as the channel length, the Doppler spread, the location of the multipath, etc. Thus, there are two problems, namely, there are too many features to describe the UWA channel, and there may be estimation errors in the features.

Different from the existing works that classify the UWA channels using the channel features, in this paper, we propose a new channel classification method that uses the BERs of all MCSs to classify UWA channels. That makes sense because we only care about the BER performance. For example, for two UWA channels with different structures, if the BERs of all MCSs are the same, we could consider the two channels to belong to the same class. If the two channels have similar structures but the BERs are different, the transmitter still should divide them into different classes. Based on this principle, we design a channel classification criterion based on the BERs of all MCSs, which is given as follows.

For channel $\boldsymbol{h}_i$ and $\boldsymbol{h}_j$, if

$$\sum_{q=1}^{Q} \gamma_q |e(q, \boldsymbol{h}_i) - e(q, \boldsymbol{h}_j)|_p \leq \epsilon, \tag{16}$$

then, $\boldsymbol{h}_i$ and $\boldsymbol{h}_j$ belong to the same channel class. Otherwise, $\boldsymbol{h}_i$ and $\boldsymbol{h}_j$ belong to different classes. $e(q, \boldsymbol{h}_i)$ represents the BER of the $q$-th MCS under the channel $\boldsymbol{h}_i$, $\epsilon$ is the threshold of channel classification, $\gamma_q$ is the weight of each MCS. $|\cdot|_p$ is the $p$-norm. Then, the channel state of $\boldsymbol{h}_i$ can be expressed by a $Q$-dimension BER vector $\boldsymbol{e}(\boldsymbol{h}_i) = [e(1, \boldsymbol{h}_i), e(2, \boldsymbol{h}_i), \cdots, e_i]$. In this way, we reduce the dimension of the channel to $Q$ dimensions.

In (16), $\gamma_q$, $p$ and $\epsilon$ have an impact on the performance. In this paper, we set $\gamma_q = 1$ and adopt $l_1$ norm. The value of $\epsilon$ affects the number of classes. According to (16), a smaller $\epsilon$ will introduce a larger number of channel classes. However, a system with a larger number of classes may not necessarily achieve better performance. The optimal $\epsilon$ can be obtained from the grid search method.

It is worth noting that the meaning of $e(q, \boldsymbol{h}_i)$ is a little bit different from $e_n$ in (11). In (11), $e_n$ is the BER that transmits the data from the transmitter node to the receiver node under the forward UWA channel. The transmission power is also a variable in (11). $e(q, \boldsymbol{h}_i)$ in (16) refers to the feedback transmission under the feedback UWA channel, which is given in the next subsection. The transmission power is fixed under this condition.

It should be pointed out that the transmission power also affects the classification performance. If the transmission power is too large that it introduces a large feedback SNR, the BERs of all channels are close to 0. In this case, $\boldsymbol{e}_i$ is no different from $\boldsymbol{e}_j$. It is not possible to perform channel classification under this condition. Thus, there also exists an optimal feedback SNR. In addition, although we actually classify the feedback channel, the transmitter can still use this information to adaptively modulate the forward channel due to the correlation between the channels.

### 4.2. Ber Estimation Based on Deep Learning Networks

Due to the lack of a mathematical model for UWA channels, there is currently no mathematical BER expression for UWA communications. Thus, it is hard to find the relationships between the channel features and BER performance based on the model-driven method. It is also the main challenge for AMC UWA communications.

In this paper, we propose an end-to-end data-driven scheme based on deep learning to find out the relationship between the channels and BERs performance. Deep learning has been successfully applied in a wide range of areas with tremendous enhancement, such as computer vision, speech recognition, natural language processing, wireless communications [39–41], etc. Compared with other machine learning algorithms, the deep learning approach can learn the features from the data directly, thus, it reduces the difficulty of preprocessing the input data.

For the deep learning network, its key works are the structure of the network, the input data, and the output data of the network. For the network structure, in this paper, we utilize fully connected deep neural networks (DNN). For the input data of the network, some works use the estimated channels [6] or the channel features [24] extracted from the channels. It implies that these methods need to estimate the channels, which may introduce estimation errors.

Different from the existing works, we directly input the received signal to the network, which avoids estimating the channel and extracting features from the UWA channels with complex and unknown models. It uses the feature extraction capabilities of deep neural networks.

It is noted that the AMC processing is implemented in the transmitter node. However, it is hard to directly obtain information about the forward channels. Thus, the BER estimation is implemented at the transmitter using the feedback link. Since the UWA channel is modeled as FSMC with unknown transition probability, if we could obtain the feedback

channel state, it could be used to generate the state of the forward channel after learning the channel transition regular.

In order to make sure the network could learn the features from the received feedback signal, we fix the pilot signal in each transmission. For the *m*-th transmission, the feedback signal is given by

$$x_m = [x^p, x^g, t_m], \tag{17}$$

where $x^p$ is the fixed pilot signal, which is the same for each feedback transmission. $x^g$ is the guard interval whose length is larger than the channel and $t_m$ is the information data that contains the previous forward transmission result. At the transmitter, the received signal $y_m$ is given by

$$y_m = x_m \otimes h_m + w_m, \tag{18}$$

where $\otimes$ represents the convolution, $h_m$ denotes the feedback UWA channel and $w_m$ denotes the noise.

Since there is a guard interval in the transmitted signal, the received signal can be divided into two non-overlapping signals $y_m^p$ and $y_m^d$, where $y_m^p$ is the received pilot signal, and $y_m^d$ is the received data signal. It is noted that $x_p$ is the same for each feedback signal and known for the transmitter. The network could learn the channel from the $y_m^p$. Thus, we could input the $y_m^p$ into the network directly.

Since most of the deep learning network algorithms implement in the real domain, the input data could be reshaped. The received signal is reshaped as

$$\Phi_m = [\Re[y_m^p]^T, \Im[y_m^p]^T]^T, \tag{19}$$

where $\Re[\cdot]$ and $\Im[\cdot]$ denote the real and imaginary parts of the matrix.

In addition to the received pilot signal, the index of MCS $q$ should be used as the input data of the network. Then, the final input data of the network is given by

$$U_m = [\Re[y_m^p]^T, \Im[y_m^p]^T, q]^T. \tag{20}$$

It is apparent that the output of the network should be the real BER $e_m$ for modulation and coding scheme $q$. As we all know, the range of BER is $[0, 1]$. However, the distribution of BER is not uniform. Most values are less than 0.1. This feature of BER will decrease the performance of the network. In order to reduce the dynamic of the BER, we adopt the log transformation to preprocess the BER data. Then, the label of the network is set as

$$\vartheta_m = -\frac{1}{\log_2 e_m}. \tag{21}$$

For the supervised learning method, we need training data to train the network. Since there is no close form BER expression for UWA channels, we can only get the BER $e_m$ through Monte Carlo simulation. In this way, we construct a training set that consists of $M$ training samples, $\{U_m; \vartheta_m\}, m = \{1, \ldots, M\}$.

We focus on the accuracy of the BER estimation. Generally speaking, there are two metrics: mean absolute error (MAE) and mean absolute percentage error (MAPE) as the loss function. Their expressions are given as

$$\text{MAE} = \frac{1}{M} \sum_{m=1}^{M} |\hat{\vartheta}_m - \vartheta_m|, \tag{22}$$

$$\text{MAPE} = \frac{1}{M} \sum_{m=1}^{M} |\frac{\hat{\vartheta}_m - \vartheta_m}{\vartheta_m}|. \tag{23}$$

Figure 4 gives the structure of the proposed DNN. It consists of an input layer, hidden layers, and an output layer. Then, we can use this network to realize the BERs estimation of all MCSs.

In conclusion, the proposed end-to-end DNN realizes the following functions: estimate the channel according to the pilot signal, capture the features of channels, and estimate BER based on the captured features. Then, based on the estimated BERs of all MCSs, we classify the feedback channel according to (16).

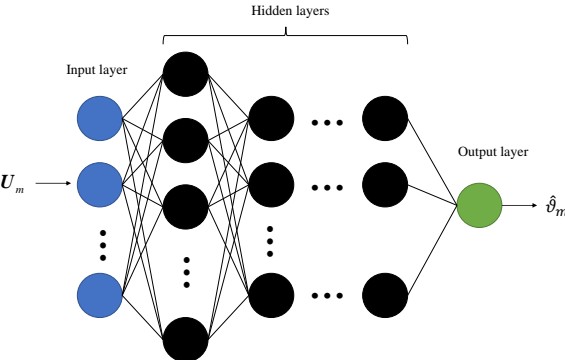

**Figure 4.** The structure of the proposed DNN for BER prediction.

## 5. Adaptive Modulation Based on DDQN Algorithm

The transmitter classifies the UWA channels according to the method mentioned in the above section. Recall our goal in (13): we want to maximize the energy efficiency by choosing optimal transmission parameters $\{r_n, M_n, P_n\}$. Since we model the time-varying channels as FSMC, problem (13) can be formulated as a Markov Decision Process (MDP). Due to the channel state transition probability and the BER expression in (11) being unknown, the traditional optimization approach or dynamic programming (DP) cannot solve the problem. Thus, we adopt the deep reinforcement learning (DRL) method to solve (13).

RL [42] is an important branch algorithm of machine learning. It consists of four key concepts: agents, states, actions, and rewards. The agent makes an action based on their state and moves from one state to another. The action is evaluated by the corresponding rewards and the agent is trained to maximize the cumulative reward. Thus, the agent learns from the interactions with its environment. Due to the lack of prior knowledge of the channels, we use a model-free RL method, Q-learning.

### 5.1. A Brief of Q-Learning

The Q-learning algorithm is an unsupervised RL algorithm, which is widely used to solve the problem where the environmental model and the transition probability are unknown. In Q-learning, the agent finds the optimal policy $\pi^*(s)$ to maximize the expected discounted long-term reward by updating the action-value function $Q(s, a)$, where $s$ is state and $a$ is action in the current step. The action-value function is updated as

$$Q(s, a) = Q(s, a) + \alpha[R + \gamma \max_{a' \in \mathcal{A}} Q(s', a') - Q(s, a)], \tag{24}$$

where $\alpha \in (0, 1)$ is the learning rate. $\gamma \in (0, 1)$ is the discount factor. $s'$ is the state of next iteration. $\mathcal{A}$ is the set of actions. $R$ is the reward for the action $a$. After enough iterations, we could obtain the optimal action-value function $Q^*(s, a)$ and the optimal policy can be expressed as

$$\pi^*(s) = \arg\max_{a \in \mathcal{A}} Q^*(s, a). \tag{25}$$

### 5.2. Ddqn

In our problem, modulation order, coding rate, and transmit power are discretized into $I$, $J$, and $K$ levels, respectively. Therefore, the dimension of the action space is $I \times J \times K$. And after channel classification, the dimension of the channel state is $Q = I \times J$. Generally speaking, Q-learning is mainly suitable for a small dimensional system. For a system with large state-action pairs, most states can rarely be visited and it is easy to be trapped in the curse of dimensionality. Besides, the Q-learning scheme is actually built on a look-up table to store all $Q(s, a)$. Thus, it also takes a long time to search for the corresponding value for a big table, and the memory may not be enough to maintain the table.

Then, to avoid this bottleneck in the Q-learning-based method, a deep Q-network (DQN) is proposed. DQN uses a train DNN to calculate the action-value function $Q(s, a; \theta)$, where $\theta$ are the parameters of the train DNN. The action-value function is updated as

$$Q(s, a; \theta) = R + \gamma \max_{a' \in \mathcal{A}} Q(s', a'; \theta), \tag{26}$$

where $a$ is selected as

$$a = \begin{cases} \text{a random action in } \mathcal{A}, \text{ with prob. } \epsilon, \\ \arg\max_{a \in \mathcal{A}} Q(s, a; \theta), \text{ with prob. } 1 - \epsilon. \end{cases} \tag{27}$$

In the DQN scheme, there is another target network with parameters $\theta^-$ to evaluate the network $\theta$. The two networks have the same structure, and the parameters of the target network $\theta^-$ are copied from the train network $\theta$ every $C$ step and kept constant at other steps.

Experience replay is another key concept of DQN. The experience data of one step $(s, a, R, s')$ is stored in the experience replay buffer and used for updating the parameters $\theta$. The parameters $\theta$ are updated based on the gradient descent algorithm to minimize the loss function. The loss function based on the mean square error (MSE) is given by

$$L(\theta) = \frac{1}{D} \sum_{t=1}^{D} \left( (R + \gamma \max_{a' \in \mathcal{A}} Q(s', a'; \theta^-) - Q(s, a; \theta) \right)^2, \tag{28}$$

where $D$ is the size of mini-batch data from the experience replay buffer.

The max operator in DQN uses the same network parameter $\theta$ to select and evaluate action. This makes an overestimation problem. To prevent this from happening, a double deep Q-network (DDQN) algorithm [43] is proposed, which uses different neural networks to decouple the selection and evaluation. In the DDQN algorithm, the action-value function is calculated by

$$Q(s, a; \theta) = R + \gamma Q(s', \arg\max_{a' \in \mathcal{A}} Q(s', a'; \theta); \theta^-). \tag{29}$$

The loss function is given by (28).

$$L(\theta) = \frac{1}{D} \sum_{t=1}^{D} \left( R + \gamma Q(s', \arg\max_{a' \in \mathcal{A}} Q(s', a'; \theta); \theta^-) - Q(s, a; \theta) \right)^2. \tag{30}$$

### 5.3. Ddqn-Based AMC

Based on the above introduction, we solve our problem (13) using the DDQN algorithm. We first define the definition of the key factors. It is apparent that the **agent** in our problem is the transmitter node. The agent chooses the optimal transmission parameters: coding rate $r_n$, modulation order $M_n$ and transmitted power $P_n$. Thus, the **action** is $a = [r_n, M_n, P_n]$. Before the $n$-th transmission, the agent first obtains the feedback signal from the receiver node. The feedback signal contains the transmission result of the last transmission $b_{n-1}$. Then, the successful transmitted data after $(n-1)$ transmis-

sions is $B_n = \sum_{l=1}^{n-1} b_l$. Besides, the agent also uses the fixed pilot to classify the feedback UWA channels, $e_{n-1}$, and saves different channels to the BER buffer. Thus, the **state** is $s = [e_{n-1}, B_n, n]$.

Our goal is to maximize the average energy efficiency after $N$ transmissions and make sure the average data rate is larger than a threshold. Then, the **reward** is defined as

$$R_n = \frac{b_n}{E_n} + \Delta, \tag{31}$$

where

$$\Delta = \begin{cases} 0, \text{ if } n < N, \\ -\omega_1(B^* - B_n), \text{ if } n = N, B_n < B^*, \\ \omega_2, \text{ if } n = N, B_n \geq B*. \end{cases} \tag{32}$$

$\Delta$ is a parameter to make sure the agent satisfies the rate demand. $B^* = TNc^*$ is the lower bound of the number of data that satisfies the minimum data rate. $\omega_1$ and $\omega_2$ are positive values.

According to the above definition, we can solve (13) by the DDQN algorithm. The detail of the algorithm is summarized in Algorithm 1. The flow chart of the entire proposed scheme is given in Figure 5.

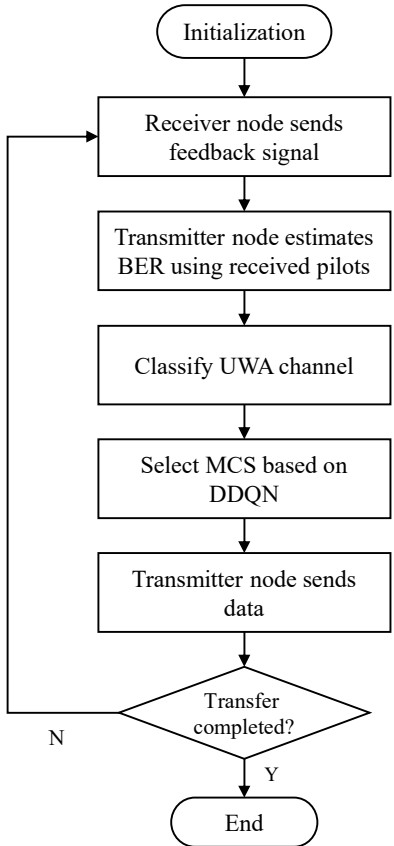

**Figure 5.** The flow chart of the proposed scheme.

---

**Algorithm 1** Proposed algorithm.

---

1: Stage 1: Training stage
2: Initialize $\mathcal{R}_c$, $\mathcal{M}$, $\mathcal{P}$, $\alpha$, $D$, $\boldsymbol{\theta}$, $\boldsymbol{\theta}^-$
3: Initialize the experience buffer as **E** and BER buffer
4: **for** Each training episode **do**
5:     Initialize $\boldsymbol{s}_1 = [\boldsymbol{e}_0, B_1 = 0, 1]$
6:     **for** Transmission episode $\boldsymbol{n} = 1, 2, \ldots$ **do**
7:         Choose action $\boldsymbol{a}_n$ via (27)
8:         Take $\boldsymbol{a}_n$ to transmit a data packet
9:         Obtain the feedback signal
10:        Calculate reward $R_n$ via (31)
11:        Estimate $e(\boldsymbol{h}_n)$ via feedback pilot
12:        **for** Each BER buffer element $\boldsymbol{e}_i$ **do**
13:           **if** (16) is satisfied **then**
14:             $\boldsymbol{e}_n = \boldsymbol{e}_i$
15:           **else**
16:             $\boldsymbol{e}_n = e(\boldsymbol{h}_n)$
17:             Add $\boldsymbol{e}_n$ to BER buffer
18:           **end if**
19:        **end for**
20:        Observe new state $\boldsymbol{s}_{n+1} = [\boldsymbol{e}_n, B_{n+1}, n+1]$
21:        Store $(\boldsymbol{s}_n, \boldsymbol{a}_n, R_n, \boldsymbol{s}_{n+1})$ in experience buffer
22:        **if E** is full **then**
23:           Randomly sample $D$ experiences from **E**
24:           Optimize $\boldsymbol{\theta}$ by minimize (30) with $\alpha$
25:        **end if**
26:        Every $C$ transmission episodes, update $\boldsymbol{\theta}^- = \boldsymbol{\theta}$
27:     **end for**
28: **end for**
29: Save network $\boldsymbol{\theta}$ and BER buffer
30: Stage 2: Test stage
31: **for** Each test episode **do**
32:     Initialize $\boldsymbol{s}_1$
33:     **for** Each transmission episode **do**
34:         Choose action $\boldsymbol{a}_n$ via (27) with $\epsilon = 0$
35:         Take $\boldsymbol{a}_n$ to transmit a data packet
36:         Obtain the feedback signal
37:         Estimate $e(\boldsymbol{h}_n)$ via feedback pilot
38:         Choose the element $e(\boldsymbol{h}_i)$ that minimizes (16)
            from the BER buffer as $\boldsymbol{e}_n$
39:         Observe new state $\boldsymbol{s}'_{n+1}$
40:     **end for**
41: **end for**

---

## 6. Simulation Results and Discussions

In this section, we evaluate the performance of the proposed AMC scheme. We consider a communication link with a transmitter node and a receiver node. The carrier frequency and bandwidth are 12.5 kHz and 5 kHz, respectively. The communication system is based on SC-FDE. The length of each block is 1024, and there are 10 blocks in each packet. Three modulation schemes, (BPSK, QPSK, and 8PSK), and three coding rates, $(2/3, 1/2, 1/3)$, are considered.

We use the statistical model in [38] to generate complex-value UWA channels. The depth of water and the depth of the transmitter node is 100 m and 20 m, respectively. The depth of the receiver node ranges from 20 m to 80 m with a step size of 0.375 m, and the minimum and maximum ranges in horizontal between the transmitter and receiver are

2500 m and 5000 m, respectively. The step size horizontal is 78.125 m. Thus, the AMC scheme has a total of 5120 UWA channels.

For the AMC scheme, the maximum tolerable BER is $\delta = 0.001$. The threshold of the channel classification is $\epsilon = 0.5$. The minimum tolerable communication rate is $c^* = 6.67$ kbps, which equals QPSK modulation with 2/3 convolution code. We do not consider the influence of the pilot and CP on the communication rate. The parameters $\omega_1$ and $\omega_2$ in the reward function are 1 and 5, respectively. The received SNR of the feedback link is about 12 dB. The total number of transmissions is $N = 15$. We assume that the transmitter node has enough energy and there is no constraint about the maximum transmission power.

We first evaluate the performance of the proposed BER estimation method. The length of the fixed pilot signal is 1280, and there are two parameters to denote the modulation order and coding rate. Thus, the number of neurons in the input layer is $1280 \times 2 + 2 = 2562$. The DNN has 4 hidden layers, and the number of neurons in each hidden layer is 128, 64, 32, and 16, respectively. The output layer only has one neuron. The rectified linear unit (ReLU) activation function is utilized for the layer except for the output layer. For the output layer, we do not use the activation function. The Adam optimizer is used for training the network. The learning rate is 0.001. For the BER estimation network, a dataset is also required to train the network. A total of 30,720 UWA channels are generated for BER estimation, in which the number of training data set, validation data set and test data set are 24,576, 3072, and 3072, respectively.

We test the performance with MAE and MAPE loss functions. Figure 6 shows the convergence of the proposed DNN for BER estimation with different loss functions. From this figure, we find that the network converged after 50 epochs for both loss functions. The MAE and MAPE in the validation data set after convergence are about 0.175 and 0.125, respectively. It is noted that the MAE and MAPE in Figure 6 are about the BER data after preprocessing, $\vartheta$. The average MAE and MAPE of the proposed method for the BER in the test data set are 0.0273 and 0.4326, respectively. Figure 7 gives a segment of estimated BER based on MAPE in the test data set. From this figure, we could see that the proposed method could estimate the BER well.

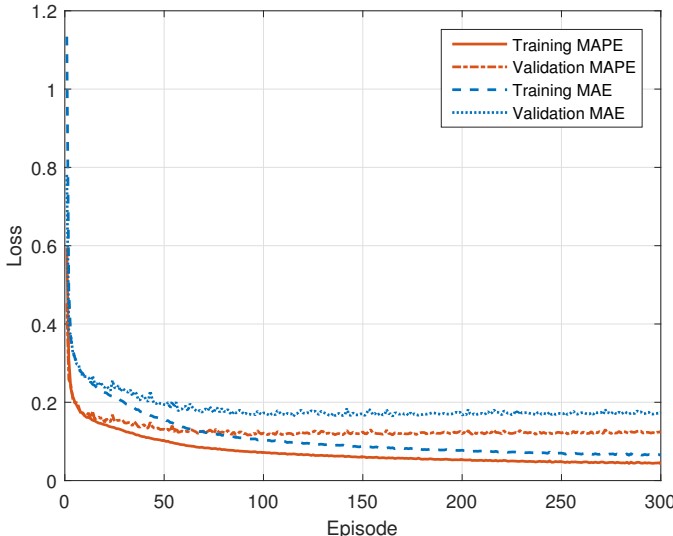

**Figure 6.** The convergence of the proposed DNN for BER estimation with different loss functions.

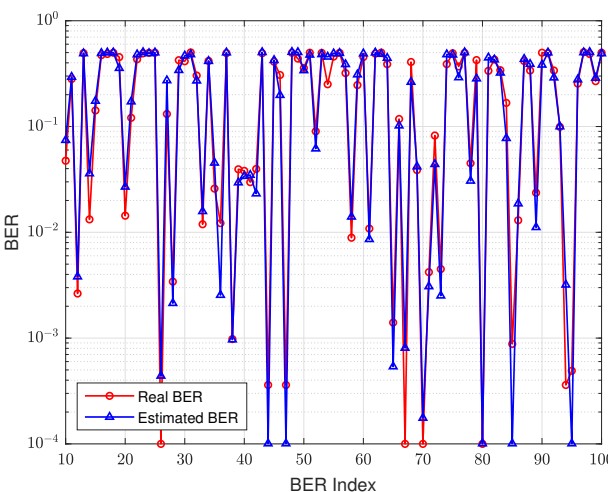

**Figure 7.** The segment of BER estimation in the test data set.

Figures 8 and 9 give the convergence and energy efficiency of the proposed DDQN scheme based on the real BER and estimated BER with MAPE loss function, respectively. The average energy efficiency is 564.23 bit/J and 556.20 bit/J based on the real BER and estimated BER, respectively. The performance gain is quite small. It is worth noting that although the accuracy of BER estimation is about $10^{-2}$ and the QoS is about $10^{-3}$, the performance of the proposed AMC scheme based on the estimated BER could still work well. This is because the channel classification adopts the $l_1$ norm of BERs and is not sensitive to the BER estimation accuracy, which means that the performance of the channel classification is robust to the BER estimation accuracy. Figure 10 gives the throughput convergence of the proposed scheme. It can be found that the throughput after convergence meets the system constraints.

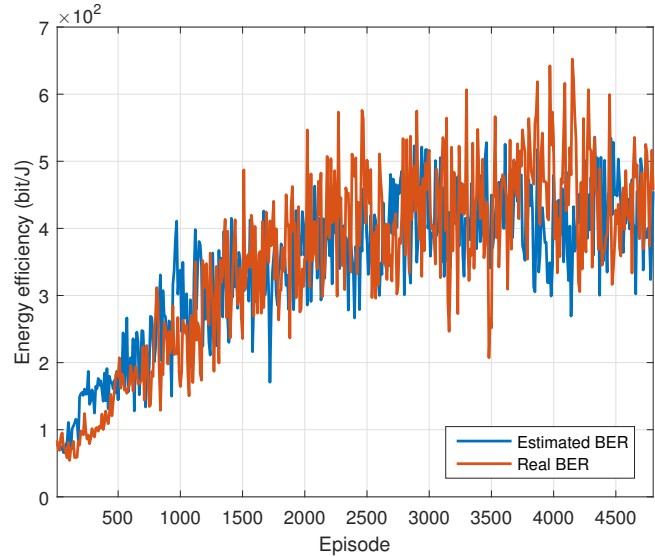

**Figure 8.** The convergence of the proposed DDQN scheme based on the real BER and estimated BER.

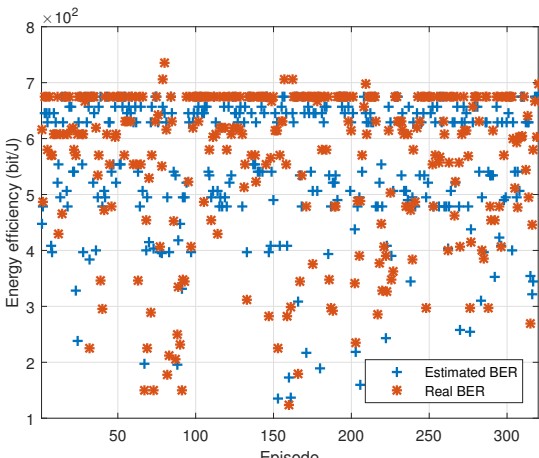

**Figure 9.** The performance of the proposed DDQN scheme based on the real BER and estimated BER.

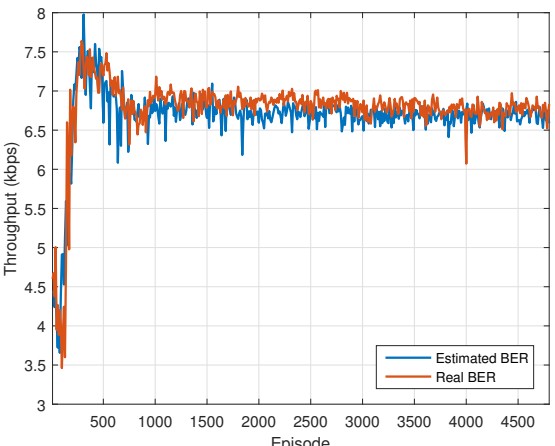

**Figure 10.** The throughput convergence of the proposed DDQN scheme based on the real BER and estimated BER.

The performance of BER estimation is related to the feedback received SNR. Thus, the SNR of the feedback link also affects the performance of energy efficiency. Figure 11 gives the performance of energy efficiency with different feedback link SNR. From the figure, we find that for the AMC system, the network based on the MAPE loss function has better performance. Thus, we finally choose the MAPE as the loss function in the AMC system. The performance does not increase all the time with increasing the feedback link SNR. That is because the feedback link SNR affects the BER of the feedback link. The transmitter classifies the feedback channel based on the estimated BERs. If the feedback SNR is too large, then all the estimated BERs will approach 0. The transmitter could not classify the channel if all BERs are 0. The transmitter is also unable to classify all channels with a BER close to 0.5, which is obtained by a small feedback SNR. Thus, there exists an optimal feedback SNR. According to Figure 11, the optimal feedback SNR is 12 dB.

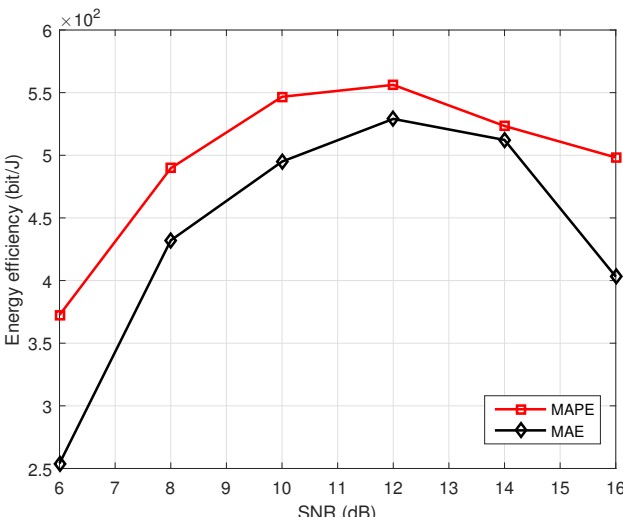

**Figure 11.** The performance of energy efficiency with different feedback SNR.

According to (16), the threshold $\epsilon$ affects the number of channel classes. A smaller threshold $\epsilon$ means more channel classes, and the number of channel classes will affect the performance. Figure 12 shows the effect of the threshold $\epsilon$ on the performance. It can be found that more channel classes do not mean having a better performance. It has the best performance with $\epsilon = 0.5$. Figure 13 gives the result of the channel state transition probability results based on the estimated BER with $\epsilon = 0.5$. From this figure, we find that all channels are divided into six classes. The color in the table represents the number of times the current channel appears.

Figure 14 shows the performance comparison of different schemes with different target throughputs. We design three comparison schemes. The first one is labeled as *Optimal AMC* in the figure, in which the transmitter node knows the channel perfectly. For each target throughput demand, the transmitter could choose the optimal modulation order and coding rate with the minimum transmission power. It can be regarded as an upper-bound performance. The second scheme is labeled as *Fixed MCS*. In this scheme, the transmitter also knows the channel perfectly and chooses a fixed MCS for each target throughput demand. The MCS could be different for different target throughput constraints. The difference between the *Fixed MCS* and the *Optimal AMC* is that under each target throughput requirement, the MCS is fixed for the *Fixed MCS* scheme, while the MCS is changed for the *Optimal AMC* scheme. The third scheme adopts the Q-learning algorithm to realize the AMC process, which is labeled as *Q-learning based AMC*. From Figure 14, we find that the performance of the proposed DDQN-based AMC scheme is near the optimal energy efficiency with perfect channel information, and it is far superior to the other two schemes. It is worth noting that the Q-learning-based scheme is also better than the fixed MCS scheme, although the fixed MCS scheme perfectly knows the channels. It shows the superiority of the proposed BER estimation and channel classification methods.

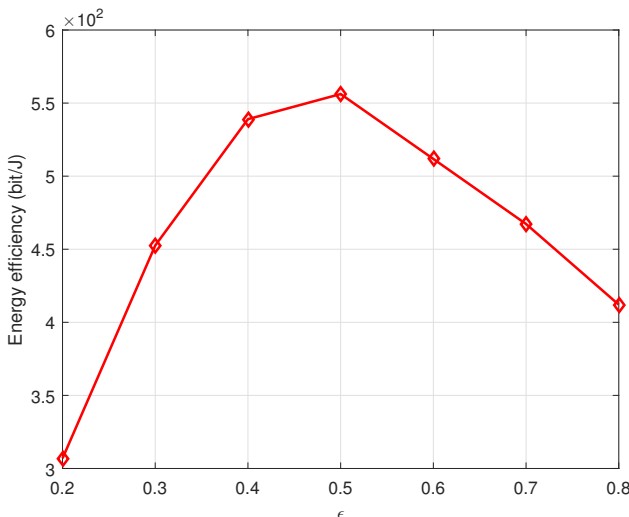

**Figure 12.** The performance of energy efficiency with different $\epsilon$ .

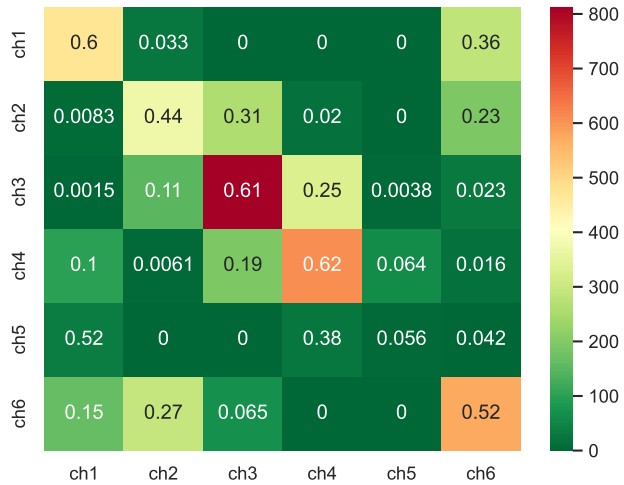

**Figure 13.** The channel state transition probability with $\epsilon = 0.5$.

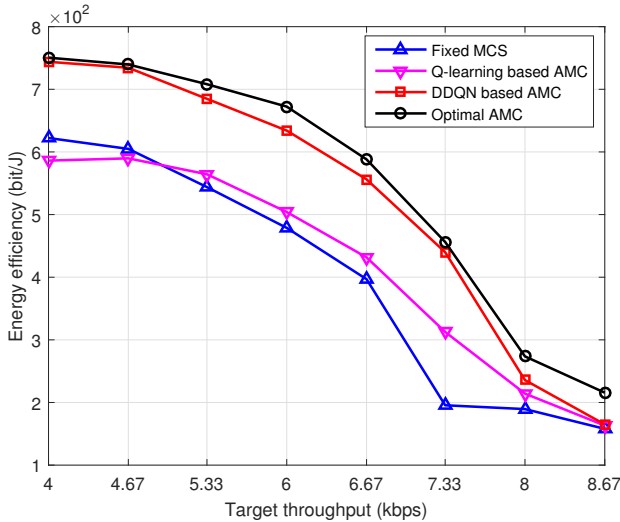

**Figure 14.** The performance comparison of different schemes with different target throughput.

## 7. Conclusions

In this paper, we have investigated the AMC problem for point-to-point UWA communication systems to maximize energy efficiency by jointly scheduling the modulation order, coding rate, and transmission power. We first proposed a BER estimation method based on a deep neural network by using a fixed pilot, which could realize channel estimation, feature extraction, and BER estimation. Then, the estimated BERs of all MCSs are used to design a channel classifications method. Additionally, the UWA channels are modeled as a finite-state Markov chain with unknown transition probability. Thus, we formulated the AMC problem as MDP and considered it in the DRL framework. Based on the result of channel classification, a DDQN-based scheme is proposed. Simulation results show the effectiveness of the proposed method. The proposed BER estimation method has a low estimation error MAPE and can accurately estimate BER. The channel classification method can reduce a large number of channels into a few classes, which significantly reduces the complexity of channel processing. On the basis of the above, the DDQN-based AMC scheme perfectly achieves the expected optimization goal, and the energy efficiency of the proposed AMC scheme is close to the optimal performance with perfect channel state information. It is far superior to Q-learning-based AMC and fixed MCS schemes.

**Author Contributions:** Conceptualization, L.J., C.D.; formal analysis, L.J.; methodology, L.J. and H.Y.; software, L.J., C.D.; validation, C.H. and H.Y.; writing—original draft, L.J.; writing—review and editing, H.Y. and C.H.; visualization, L.J., C.D.; supervision, W.S. and H.Y.; project administration, L.J., H.Y.; funding acquisition, L.J. All authors have read and agreed to the published version of the manuscript.

**Funding:** This research was funded by the National Natural Science Foundation of China (61801079, 62071383, 61871418), the open research fund of the State Key Laboratory of Integrated Services Networks (ISN22-15), the Science and Technology on Underwater Information and Control Laboratory (6142218200408), and the Fundamental Research Funds for the Central Universities (DUT22JC05).

**Data Availability Statement:** The data presented in this paper are available after contacting the corresponding author.

**Acknowledgments:** The authors would like to thank the anonymous reviewers for their careful reading and valuable comments.

**Conflicts of Interest:** The authors declare no conflict of interest.

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
