# Peer review of "Adaptive Modulation and Coding for Underwater Acoustic Communications Based on Data-Driven Learning Algorithm"

_remotesensing, doi:10.3390/rs14235959_

Round 1

Reviewer 1 Report

Please to kindly see the attached file. Thank you.

Author Response

1. I miss some graphic or scheme that present the finite-state Markov chain (FSMC) used in this work. I suggest including it in the Section 5, where is very convenient for understanding the selection of the modulation.

Response: Thanks for your suggestion. We have added some explanation about the FSMC in Section 3 of the revised manuscript as your suggested.

2. The energy efficiency presented in Figure 10, it depends on the time? I mean, if the number of transmissions/receptions is increased, can the efficiency change? I make this comment to see if the value of ε=0.5 is constant or can change with the link conditions.

Response: Thanks for your comment. The energy efficiency is shown in Figure. 10 is dependent on the number of channel types in the training phase, independent of time. When the number of transmission/reception increases, more iterations are required for algorithm convergence, but the final energy efficiency is not affected. The classification threshold ε needs to be adjusted to an appropriate size to match the SNR of the feedback link, and then fixed. Because in the training phase, the classification threshold ε determines the number of channel types, which will lead to changes in the performance of the algorithm. Specifically, as shown in equation (16), when ε is too large, it is considered one channel type as long as the BER vector (represents the channel) gap does not reach ε. The transmitter node will use one MCS transmission, even if the channels need to use different MCS for optimal performance. When ε is too small, subtle BER vector differences will be considered as different channel types, and then the transmitter node will use multiple MCS for transmission, even if these channels only need to use the same MCS. Obviously, either case is inappropriate and will result in reduced performance.

3. “ simulation” better in capital letter: “6. Simulation”

Response: Thanks for your suggestion. We have modified it in the revised manuscript.

4. Figure 1: Although there are many variations for the UWA channel, I think is necessary to include some range of the parameters used (e.g. Temperature, Salinity, …) in a legend box or in the caption.

Response: Thanks for your suggestion. We have modified it as you suggested in the revised manuscript.

Reviewer 2 Report

1.In abstract, bit error ratio (BER) estimation method  based on deep neural networks (DNN). Is this can be realized in node, for the cost of  complex processing and power consuming.

2. In the content, The transmitter node first uses the received pilot signal of the feedback link to estimate the BERs of all MCSs. If this need all BERs of all MCSs, then how about the consumption of pilot ?

3. As in context , A fixed  pilot signal is transmitted in each transmission, and the received pilot signal is directly used as the input of the DNN. How the received pilot signal was sent back to transimittor ?

4. Fig1, the condition of simulation should be mentioned , and maybe classified with different factors more efficiently .

5.For the part of Underwater acoustic channel model , that should be some more deep anayslzed  such as space-time variant , phase distortion, etc, other than only intensity .

6.   However, it is hard to directly use the channel  state information due to the complexity of UWA channels, even the channel impulse  response is well estimated. In the field of research , there exists many criteria of UWA channel.Using  BER as criteria to classify channel , it maybe affect by characteristics of pilot data itself.

7. Adaptive modulation based on DDQN algorithm, it was curious that the calculation consumption and cost of processing volume. 

8. For the conclusion part , it need some supplement for the merit and consumption of method presented here.

9. It suggest to provide a flow chart of algorithm to be more clear for readers.

Author Response

1. In abstract, bit error ratio (BER) estimation method based on deep neural networks (DNN). Is this can be realized in node, for the cost of complex processing and power consuming.

Response: Thanks for your comment. The deep neural network has been widely used. Considering cost and computing power, FPGA-based deep neural networks have outstanding performance.

For instance, [1] implemented CNN-based natural scene recognition on FPGA, and [2] implemented real-time face recognition. For low power requirements, [3] FPGA-based hand pose estimation algorithm is 4.2 times faster and 577.3 times more energy efficient than GPU-based. The natural language processing acceleration framework proposed by [4] can compress the model to 1/16 of the original size, achieve 27.07-81 times performance improvement, and is 8.8 times more energy efficient than CPU-based. In conclusion, applying deep neural networks on underwater acoustic communication nodes is a very promising and low-cost scheme. We have added some explanations in the revised manuscript.

2. In the content, the transmitter node first uses the received pilot signal of the feedback link to estimate the BERs of all MCSs. If this need all BERs of all MCSs, then how about the consumption of pilot?

Response: Thanks for your comment. Only one pilot signal is needed to estimate the BER of all MCSs. Because the pilot signal is fixed, the received pilot is only related to the UWA channel and transmit power and has nothing to do with the specific MCS. In section IV B, when generating the data set, pilots xp and data tm are transmitted together using different MCS, and then the BER of received pilots yp and tm is obtained. For a channel, since the transmit power is fixed, the BER of tm will be different under different MCS, while the received pilot frequency yp is the same. We obtained the BER estimation model using the received pilots yp and MCS as training data and the BER of tm as the training label. It only needs to input a received pilot and an MCS to estimate the corresponding BER. Further, a pilot combined with all MCSs can estimate the BER of all corresponding MCSs.

3. As in context, A fixed pilot signal is transmitted in each transmission, and the received pilot signal is directly used as the input of the DNN. How the received pilot signal was sent back to transmitter?

Response: Thanks for your comment. As mentioned at the beginning of section III, both the transmitter node and the receiver node can transmit. As described at the end of section III C, the receiver node sends a feedback signal to the transmitting node, which contains a fixed pilot signal and the result of the previous transmission. Hence the received pilot signal is that the transmitter node has received the pilot signal from the receiver node, it has reached the transmitter and does not need to be sent back to the transmitter again.

4. Fig1, the condition of simulation should be mentioned, and maybe classified with different factors more efficiently.

Response: Thanks for your suggestion. We have added the explanation about the simulation in section 3.2 in the revised manuscript.

5. For the part of Underwater acoustic channel model, that should be some more deep analyzed such as space-time variant, phase distortion, etc., other than only intensity.

Response: Thanks for your suggestion. We have added some analysis about the UWA channel in section 3.2 in the revised manuscript.

6. However, it is hard to directly use the channel state information due to the complexity of UWA channels, even the channel impulse response is well estimated. In the field of research, there exists many criteria of UWA channel. Using BER as criteria to classify channel, it maybe affects by characteristics of pilot data itself.

Response: Thanks for your comment. The main idea of BER-based classification criteria is that channels with no significant difference in transmission performance should belong to the same class of channels. In addition, representing the channel as a BER vector can greatly reduce the channel state, enabling reinforcement learning methods to effectively learn the channel variation regulation.

Since the received pilot data contains UWA channel information, we use the received pilot to estimate BER based on the deep learning method. While deep learning has a strong fitting ability and is capable of handling complex UWA channels [5-7]. To improve the BER estimation ability, we also preprocess the dataset. The BER estimation results of the test set and the energy efficiency results all demonstrate the effectiveness of the proposed method.

7. Adaptive modulation based on DDQN algorithm, it was curious that the calculation consumption and cost of processing volume. 

Response: Thanks for your comment. The proposed DDQN-based adaptive modulation is an offline algorithm, which means that we can train the adaptive model when idle, and deploy it to the communication nodes after convergence. This saves a lot of time costs. For the calculation cost, we tested it on i9 9900KF, RTX2080TI. In the training phase, the proposed algorithm will increase the computational load (time cost) by 18% per training episodes. After the training is completed, since there is no need to iterate and update the DNN, the proposed algorithm will increase the computational load by 5% on the test set, which is equal to the cost when applied to the communication node. Concomitantly, there is a substantial improvement in energy efficiency, which is 24% higher than the Q-learning method and 31% higher than the fixed MCS.

8. For the conclusion part, it needs some supplement for the merit and consumption of method presented here.

Response: Thanks for your suggestion. We have added some supplements as you suggested in the revised manuscript.

  1. It suggests to provide a flow chart of algorithm to be clearer for readers.

Response: Thanks for your suggestion. We have added a flow chart in section 5 in the revised manuscript.

[1] Y. Ma, Y. Cao, S. Vrudhula, and J.-S. Seo, “Optimizing loop operation and dataflow in FPGA acceleration of deep convolutional neural networks,” in Proc. ACM/SIGDA Int. Symp. Field-Programmable Gate Arrays, pp. 45--54, Feb. 2017.

[2] Z. A. Soomro, T. Din Memon, F. Naz and A. Ali, “FPGA-based real-time face authorization system for electronic voting system,” in Proc. International Conference on Computing, Mathematics and Engineering Technologies, Sukkur, Pakistan, Jan. 2020.

[3] M. R. Al Koutayni et al., “Real-time energy efficient hand pose estimation: A case study,” Sensors, vol. 20, no. 10, pp. 2828, May 2020.

[4] B. Li et al., “Ftrans: Energy-efficient acceleration of transformers using FPGA,” in Proc. ACM/IEEE Int. Symp. Low Power Electron. Des., pp. 175--180, Aug. 2020.

[5] D. Li-Da, W. Shi-Lian, and Z. Wei, “Modulation Classification of Underwater Acoustic Communication Signals Based on Deep Learning,” in Proc. MTS/IEEE Oceans, Kobe, Japan, May 2018.

[6] L. Liu, L. Cai, L. Ma, and G. Qiao, “Channel state information prediction for adaptive underwater acoustic downlink OFDMA system: Deep neural networks based approach,” IEEE Trans. Veh. Technol., vol. 70, no. 9, pp. 9063-9076, Sept. 2021.

[7] M. Zhou, J. Wang, H. Sun, J. Qi, X. Feng, and H. Esmaiel, “A novel DNN based channel estimator for underwater acoustic communications with IM-OFDM,” in Proc. IEEE International Conference on Signal Processing, Communications and Computing, Macau, China, Aug. 2020.

Round 2

Reviewer 2 Report

1. For the consumption of power, 'Recently, machine learning (ML) technology has found wide applications in broad fields, including image/audio processing, economics, and computational biology.’. It was not so convincible , especially for that underwater situation scenario, and the purpose of this research is for energy efficiency. It maybe better analyze this directly from computation cost .

Author Response

Response: Thanks for your comment. The chips of machine learning have already been used in scenarios with strict energy efficiency requirements, as shown in R[1-3]. In R[1], it shows FPGA-based hand pose estimation algorithm is 4.2 times faster and 577.3 times more energy efficient than GPU-based for low power requirements. The natural language processing acceleration framework proposed by R[2] can compress the model to 1/16 of the original size, achieve 27.07-81 times performance improvement, and is 8.8 times more energy efficient than CPU-based. In R[3], the authors realized a Stochastic Computing based LSTM in FPGA, which can reduce 73.24% energy cost compared to baseline binary LSTM. Thus, machine learning technology could be used in real-world underwater acoustic communications systems.

R[1] M. R. Al Koutayni et al., “Real-time energy efficient hand pose estimation: A case study,” Sensors, vol. 20, no. 10, pp. 2828, May 2020.

R[2]. B. Li et al., “Ftrans: Energy-efficient acceleration of transformers using FPGA,” in Proc. ACM/IEEE Int. Symp. Low Power Electron. Des., pp. 175–180, Aug. 2020.

R[3] G. Maor, X. Zeng, Z. Wang, and Y. Hu, "An FPGA implementation of stochastic computing-based LSTM," 2019 IEEE 37th International Conference on Computer Design (ICCD), 2019, pp. 38-46.